# How Social Influence Promotes the Adoption of Mobile Health among Young Adults in China: A Systematic Analysis of Trust, Health Consciousness, and User Experience

**DOI:** 10.3390/bs14060498

**Published:** 2024-06-14

**Authors:** Jianfei Cao, Hanlin Feng, Yeongjoo Lim, Kota Kodama, Shuo Zhang

**Affiliations:** 1College of Business Administration, Ritsumeikan University, 2-150 Iwakura-cho, Ibaraki 567-8570, Osaka, Japan; cjf869419482@gmail.com (J.C.); lim40@fc.ritsumei.ac.jp (Y.L.); 2Graduate School of Technology Management, Ritsumeikan University, Ibaraki 567-8570, Osaka, Japan; gr0328ee@ed.ritsumei.ac.jp; 3Faculty of Data Science, Nagoya City University, Nagoya 467-8501, Aichi, Japan; kkodama@fc.ritsumei.ac.jp; 4School of Management, Harbin Institute of Technology, Harbin 150006, China

**Keywords:** mobile health, health behavior, trust, health consciousness, mHealth user experience, young adults, China

## Abstract

As mobile health (mHealth) offers several advantages in healthcare, researchers are exploring the motivational factors for its adoption. However, few studies have elucidated the complex relationship between social influence and behavioral intentions to adopt mHealth among young adults in China. This study explored the impact of social influence on young adults’ behavioral intentions to adopt mHealth, the mediating roles of trust and health consciousness, and the moderating effect of mHealth user experience on the relationship between the predictors. In total, 300 valid responses were collected from a university in China, and a research model was developed. The partial least squares structural equation modeling method was used to verify the relationship between the main research variables. mHealth adoption behavioral intentions among young adults were significantly positively impacted by social influence; it indirectly increased mHealth adoption behavioral intentions by positively affecting trust and health consciousness. mHealth use weakened the positive impact of social influence on trust and health consciousness, while user experience positively moderated the relationship between health consciousness and behavioral intentions. Trust and health consciousness play important roles in the complex multivariate relationships between social influence and behavioral intentions to adopt mHealth. Future research should consider the moderating role of the mHealth user experience. These findings enrich the mHealth technology acceptance theory framework and provide specific guidance strategies for marketing mHealth applications.

## 1. Introduction

In recent years, the advantages of mobile health (mHealth) have received attention from researchers worldwide [1]. In 2011, the World Health Organization (WHO) defined mHealth as “medical and public health practices supported by mobile devices, such as mobile phones, patient monitoring devices, personal digital assistants (PDAs), and other wireless devices” [2]. With the development of information technology and the popularity of smartphones, various services have been developed for mobile health. These include online doctor appointments as well as remote diagnosis and treatment services during patient consultations [3,4,5]; various wearable devices for monitoring and collecting patient data during consultations and post-diagnosis [6,7]; applications (apps) for managing patient medication compliance during recovery [8,9]; and apps guiding healthy eating and exercise in daily life [10,11]. Given the powerful data analysis capabilities of mobile technology and the advantage of accessing medical services anytime and anywhere, mHealth has played an essential role in the development of the medical care field, from medical resource allocation to personalized medical services [12,13].

Integrating mHealth into the traditional healthcare system implies the involvement of third-party organizations, such as cloud platform providers, users, and healthcare providers [14]. Users may be concerned about safeguarding the privacy of their personal health information [15,16]. Additionally, users must own a mobile device, such as a smartphone, to use mHealth [17], and some specific health monitoring services may require wearable devices, such as smartwatches [18,19]. Therefore, mHealth faces problems of high usage conditions and high learning costs. Despite the significant advantages of mHealth in developing the healthcare industry, its widespread adoption in society still faces many challenges [14].

Previous studies have frequently considered how older adults can benefit from mHealth [20,21,22,23]. As they are often experiencing declines in their health, the health management services provided by mHealth can be specifically useful to this population [24]. However, older adults do not use mHealth services at ideal rates [25]. Compared to young adult users, older adults tend to spend more time and energy learning information technology-related knowledge [26].

In contrast, previous research has suggested that young adults are key drivers of mHealth adoption [27,28,29]. Users of mobile healthcare applications in China tend to be younger and have high education levels [30]. Young adults constitute the largest group of Internet users in China [31]; they spend more time using the internet and have higher smartphone usage [31]. Accordingly, they have fewer technological barriers and can more easily access health-related information. In addition, openness is also a critical factor influencing willingness to use mHealth. Individuals with higher openness are more likely to absorb new ideas and accept the changes brought about by new technologies [32]. Therefore, young adults are considered a critical link in promoting mHealth adoption. One survey showed that increasing numbers of young adults are helping their family members create health records to access online medical services [33]. To encourage greater adoption of mHealth among young adults, it is necessary to conduct an in-depth exploration of the factors influencing the acceptance of mHealth.

### 1.1. Theoretical Background and Hypotheses

#### Theoretical Framework Based on Social Influence

We constructed our research framework based on theories from several domains. First, we established the foundational framework of our study based on the Unified Theory of Acceptance and Use of Technology (UTAUT) developed by Venkatesh et al. [34]. The UTAUT evolved from eight theories explaining information system user behavior, including the theory of reasoned action, technology acceptance model, motivational model, theory of planned behavior, combined theory of planned behavior/technology acceptance model, model of personal computer use, diffusion of innovations theory, and social cognitive theory, and has been found to have superior explanatory power. In the UTAUT, performance expectancy, effort expectancy, facilitating conditions, and social influence directly determine a user’s behavioral intention. Although UTAUT has demonstrated strong explanatory power within the mHealth domain [35,36], previous research indicates the necessity of delving deeper into its influential pathways to enhance its suitability for mHealth. For instance, Wang et al. amalgamated the Task-Technology Fit model (TTF) with UTAUT [37]. The model explained 68.0% of the variance in behavioral intention. To enrich the mHealth technology acceptance theory, in this study, we focused on the impact of social influence on the behavioral intention to adopt mHealth among young adults in China.

Social influence is how consumers perceive that other people who are important to them (e.g., family and friends) believe they should use a particular technology [34,38]. Many studies have confirmed that social influence positively affects behavioral intention to adopt mHealth [39,40,41]. However, this relationship is not absolute. For example, a study by De Veer et al. [42] surveyed 1014 Dutch individuals and found that social influence does not significantly affect the intention to use eHealth [42]. Moreover, a systematic literature review of the UTAUT2 showed that only about half of the studies reported a significant impact of social influence on the intention to use new technologies [43]. Therefore, we had to consider the possibility of a complex interplay involving other variables between social influence and behavioral intention to adopt mHealth services.

In the context of mHealth, Duarte and Pinho [44] suggest that adopting mHealth is a complex phenomenon. In the theoretical exploration of mHealth adoption behaviors, it is necessary to reference technology acceptance theories (such as UTAUT and UTAUT2) and consider other dimensions related to mHealth [44]. Cao et al. [28] combined the social aspects of mHealth with subjectively perceived psychological dimensions and personal characteristics in a study of young Japanese adults. They found that social influence could indirectly affect behavioral intentions to adopt mHealth through subjective psychological dimensions. However, to our knowledge, studies have not examined the complex relationship between social influence and behavioral intention to adopt mHealth among young adults in China. To fill this research gap, our study developed a social influence model to investigate mHealth adoption intentions among young Chinese adults. Specifically, we propose Hypothesis 1 based on the UTAUT.

**Hypothesis 1** **(H1).**
*Social influence positively affects behavioral intention toward mHealth.*


Building on this, we introduce two mediating variables: trust and health consciousness. Additionally, mHealth user experience was added as a moderating variable. Figure 1 illustrates the proposed model. In the following sections, we elaborate on the theoretical basis for introducing the variables and establishing the model.

### 1.2. The Introduction of Trust as a Mediating Variable

Trust is widely utilized in extended research on technology acceptance theories [28,45]. However, the definition of trust is inconsistent due to disciplinary differences [46]. Therefore, we first provide a precise definition of trust within the mHealth domain. We adopted the definition of trust from the integrative model of organizational trust proposed by Mayer et al. [47]. This theory synthesizes common characteristics across different disciplines and research contexts, including the characteristics of the trustor, trustee, and role of risk. It defines trust as “the willingness of a party to be vulnerable to the actions of another party based on the expectation that the other will perform a particular action important to the trustor, irrespective of the ability to monitor or control that other party”.

In the mHealth domain, Sheppard suggested that trust in mHealth should be considered comprehensively from the institutional, interpersonal, and technological aspects [48]. With the rapid development of mHealth, the number of mHealth apps is increasing. The personal data generated by using mHealth apps is often highly sensitive. However, according to a survey by Lewis and Wyatt, only 10% of apps provide a privacy policy (n = 600) [49]. Currently, regulatory policies related to mHealth are in a stage of continuous improvement [50]. Under such circumstances, the legal guarantee of user safety is a key factor in establishing trust [48]. Moreover, because mHealth relies on applications and smart devices developed by technology developers, accessing mHealth services means users need to accept surveillance of their personal health data [51]. This may raise ethical concerns about the extent of user surveillance acceptance. Therefore, it is crucial to consider whether users trust the security, privacy, and reliability of the services provided by technology developers [14]. Combining these theoretical backgrounds, trust in this study is defined as the degree to which users (trustors) trust relevant scientific researchers, government decision-makers, and technology developers (trustees) in the context of mHealth (role of risk).

Trust can help users overcome and accept the perceived risks associated with the uncertainty of new technologies [52]. In previous research, trust has played a crucial role in predicting behavioral intentions to adopt emerging information technologies [53]. For example, trust positively affects users’ willingness to use mobile payments and online shopping [54]. Trust has also been proven to positively influence behavioral intentions to adopt mHealth [28]. However, few studies have assessed social influence as a determining factor in establishing trust. From the diffusion of innovation perspective, when people lack direct experience with new technologies, they build their initial cognition and familiarity based on secondary knowledge, personal impressions, and other cues [55]. Cognitive familiarity also determines the establishment of initial trust in the trustee. Trust in new technology by individuals known to the trustor plays a significant role in helping establish initial trust and may influence subsequent stages [56]. In the information systems domain, social influence was found to have the most significant impact on trust beliefs [53]. Neuroscience has discovered that most choices can influence an individual’s trust preferences [57]. Based on this theoretical background, we propose the following hypotheses:

**Hypothesis 2** **(H2).**
*Social influence will positively affect trust regarding mHealth.*


**Hypothesis 3** **(H3).**
*Trust will positively affect behavioral intention regarding mHealth.*


### 1.3. The Introduction of Health Consciousness as a Mediating Variable

Health consciousness is the degree to which health concerns are integrated into an individual’s daily activities [58]. Jayanti and Burns [58] introduced health consciousness as an observational factor in an empirical study on preventive healthcare behavior. They found a significant positive relationship between health consciousness and preventive healthcare behavior. From a psychological perspective, health consciousness, as an essential personal attribute, is closely related to health motivation [59]. Individuals with high health consciousness are more likely to pay attention to health issues and implement health-promoting behaviors to avoid being unhealthy [60], such as actively obtaining health-related information [61] and consuming healthy foods [62]. Therefore, these individuals are considered excellent customers for health-related products and services [63].

Health consciousness plays an essential role in predicting behaviors towards mHealth usage [64]. Cho et al. [65] discovered that health consciousness directly influences the extent to which people adopt mHealth applications; this relationship has also been mediated by perceived usefulness [28]. Guo et al. [66] found that health consciousness strengthened the positive influence of social media on attitudes towards mHealth usage. Nevertheless, previous studies have focused only on health consciousness as a direct observational factor in mHealth adoption behavior, overlooking a crucial antecedent: environmental factors. Bandura’s social cognitive theory (SCT), proposed by Bandura, elucidates a triadic dynamic relationship among individuals, the environment, and behavior [67]. This theory suggests that the surrounding environment influences an individual’s cognition and that people acquire new behaviors by observing and imitating others. SCT has been widely applied in health promotion practices, such as promoting a healthy diet [68], smoking cessation [69], and HIV prevention [70]. Therefore, in the mHealth domain, it is reasonable to assume that mHealth usage recommendations from people around the individual can evoke their attention to their health, encouraging their positive adoption of mHealth. The specific hypotheses were as follows:

**Hypothesis 4** **(H4).**
*Social influence will positively affect health consciousness.*


**Hypothesis 5** **(H5).**
*Health consciousness will positively affect behavioral intention.*


### 1.4. The Moderating Effect of mHealth User Experience

In this study, mHealth user experience is a binary variable (yes or no), defined as whether an individual has experience using mHealth, with “no” coded as 0 and “yes” as 1. Once individuals have experience using a new technology, they may evaluate it based on their personal experiences rather than solely relying on others’ opinions [71]. If social influence operates by acquiring information from the external environment to reduce uncertainty, the actual mHealth user experience forms a definitive perception of the user experience. Therefore, we hypothesized that mHealth use experience would reduce the impact of social influence by focusing more on individual subjective awareness. The specific hypotheses were as follows:

**Hypothesis 6a** **(H6a).**
*The mHealth user experience negatively moderates the impact of social influence on behavioral intention.*


**Hypothesis 6b** **(H6b).**
*The mHealth user experience negatively moderates the impact of social influence on health consciousness.*


**Hypothesis 6c** **(H6c).**
*The mHealth user experience negatively moderates the impact of social influence on trust.*


**Hypothesis 6d** **(H6d).**
*The mHealth user experience positively moderates the impact of trust on behavioral intention.*


**Hypothesis 6e** **(H6e).**
*The mHealth user experience positively moderates the impact of health consciousness on behavioral intention.*


## 2. Methods

### 2.1. Questionnaire Design

We designed a questionnaire based on the variables introduced in this study to validate the research model and the proposed hypotheses. All question items in the questionnaire were derived from scales validated in previous studies. The items for social influence and behavioral intention were taken from the structures used in the UTAUT model proposed by Venkatesh et al. [38]. The items for the trust were adapted from research on genetically modified food by Farid et al. [45] and modified to fit the context of mHealth research. The items for HC were obtained from Guo et al. [66]. The questionnaire was translated into Chinese by three native Chinese-speaking researchers from various fields. A pre-test was conducted with 20 university students to ensure that the questions were correctly understood, leading to the removal of redundant items (HC2). This step enhanced the validity of the questionnaire. This study employed a 7-point Likert scale, where “1” represents strongly disagreeing and “7” represents strongly agreeing. The details of the measurement constructs can be found in Appendix A.

### 2.2. Data Collection

The data collection process adhered to the Ritsumeikan University Research Ethics Guidelines. This study surveyed university students in China to understand young Chinese adults’ perspectives on mHealth. In this study, we adopted the same definition as previous research, defining young adults as individuals aged 18 to 40 [72,73]. Before administering the questionnaire, participants were recruited through the WeChat social platform. Participants needed to meet the following inclusion criteria: (1) Participants were enrolled students to exclude the influence of other occupational groups on the study results; (2) participants’ ages fell within the range of 18 to 40 years old. On 12–13 April 2023, the participants completed the questionnaires in a meeting room on their campus. Those unable to visit for personal reasons were invited to participate online through the Tencent Conference. An online questionnaire was created using the WJX platform [74], and participants could complete it by scanning the QR code displayed during the meeting. An informed consent form was added to the first page of the questionnaire, to which the respondents had to agree before proceeding. Considering that respondents might have encountered the term mHealth for the first time and did not clearly understand its concept, we briefly introduced mHealth and its current applications before they filled out the questionnaire. The purpose was to assist respondents in clearly knowing whether they had any experience using mHealth services. A total of 337 responses were collected. Among these, 37 responses were excluded because they were submitted multiple times. After filtering duplicate responses, 300 valid responses were collected, resulting in an effective response rate of 89.0%. Each participant received a reward of ten CHY (approximately 1.5 USD).

The respondents’ demographic characteristics are listed in Table 1. The respondents were primarily young, with those aged 20 years and below forming the major sample group, accounting for 67.7% (203/300) of the total sample. There were 93 individuals aged 21–23, making up 31% of the sample. All respondents were students enrolled at the university, with a higher proportion of women (191/300, 63.7%). Most of the respondents had extensive experience using smartphones. Among them, 22.3% (67/300) indicated that they had experience using mHealth services.

### 2.3. Data Analysis

First, we conducted a descriptive statistical analysis of the sample using SPSS (v. 27) to identify the demographic characteristics of the participants. The model’s convergence, reliability, and discriminant validity were checked using SmartPLS (v. 4.0.9.5). Partial least squares structural equation modeling (PLS-SEM) was employed to test the significance of the paths within the model. Previous studies have widely used PLS-SEM because of its powerful predictive capabilities [75]. Compared with covariance-based structural equation modeling (CB-SEM), PLS-SEM does not require a normal distribution [76]. It is suitable for model predictions based on small samples and is more applicable to models with formative measurements [77]. Hair and Ringle [77] recommend using the “10 times rule” to estimate the minimum sample size when analyzing research models with PLS, which means the sample size should be more than ten times the maximum number of internal or external model links pointing to any latent variable in the model. Therefore, this study’s sample size of 300 is considered to meet the minimum sample size requirements. The model fit was tested using the standardized root mean square residual (SRMR).

## 3. Results

### 3.1. Measurement Model

It was necessary to assess the structure of the model before testing the hypotheses to obtain valid results. First, the factor loadings of the questionnaire items within the latent variables were evaluated to assess the model’s convergent validity. Table 2 shows that all item loadings were above 0.7 with *p* < 0.001, suggesting good convergent validity [78]. All the items were considered usable. Additionally, the variance inflation factor (VIF) for all items was below the recommended threshold of 10 [79], indicating no multicollinearity issues with the items used.

The model’s reliability was assessed using Cronbach’s alpha and composite reliability. The recommended threshold for Cronbach’s alpha and composite reliability in the PLS-SEM models is 0.7 [80]. Table 3 shows the results of the model reliability assessment, with Cronbach’s alpha ranging from 0.920 to 0.962 and composite reliability ranging from 0.943 to 0.975. Both metrics exceed the recommended threshold, thereby confirming the reliability of the model.

We assessed the model’s discriminant validity using the Fornell and Larcker criterion, which stipulates that the correlation between constructs should be less than the square root of each construct’s average variance extracted (AVE) [81]. Table 4 presents the results based on the Fornell and Larcker criteria, confirming that the model’s discriminant validity was within acceptable limits.

### 3.2. Results of the Structural Model and Hypothesis Testing

The hypotheses of this study were tested by analyzing structural equations. We calculated the path coefficients for each relationship using the bootstrapping method with 5000 samples in SmartPLS. The relationships in the model were validated in two phases. In the first phase (Model 1), we considered social influence as the independent variable, behavioral intention as the dependent variable, and trust and health consciousness as mediating variables. We found a significant positive relationship between social influence and behavioral intention (β = 0.407, *p* < 0.001), supporting H1. This means more social influence can significantly increase people’s behavioral intention to adopt mHealth. The paths connected through the mediating variables trust and health consciousness also showed statistical significance, supporting H2, H3, H4, and H5. This means that when people feel more social influence, it will indirectly affect their behavioral intention to adopt mHealth by enhancing their health consciousness and perceived trust in mHealth. The results are summarized in Table 5.

In the second phase (Model 2), we introduced the moderating variable of mHealth user experience into the model’s significant paths to examine the effect of the moderating variable on the relationships. As shown in Figure 2 and Table 5, mHealth user experience negatively moderated the impact of social influence on trust (β = −0.373, *p* < 0.001) and health consciousness (β = −0.537, *p* < 0.001). This means that after people have used mHealth, the impact of social influence on health consciousness and perceived trust in mHealth will significantly decrease. At the same time, mHealth user experience positively moderated the impact of health consciousness on behavioral intention (β = 0.245, *p* < 0.05), which means for people who have mHealth user experience, their adoption of mHealth is more likely to be driven by health consciousness, supporting H6b, H6c, and H6e. Furthermore, no significant moderating effects were found on the other paths, thus rejecting H6a and H6d. The R-squared value for behavioral intention was 0.828, indicating the high explanatory power of this model. The SRMR for Model 2 was 0.041, which was well below the recommended threshold of 0.08 [77], suggesting a good model fit.

## 4. Discussion

### 4.1. Principal Findings

This study provides an in-depth exploration of how social influences affect the behavioral intention to adopt mHealth among young Chinese adults, yielding several significant findings. First, social influence, trust, and health consciousness had a direct positive impact on behavioral intention to adopt mHealth (supported by H1, H3, and H5), which was consistent with previous research [28,82]. Second, social influence positively affects behavioral intention through trust (supported by H2 and H3). Recommendations from others to use mHealth help young adults establish trust in mHealth-related information, thereby increasing their behavioral intention to adopt mHealth. Additionally, social influence positively affected mHealth behavioral intentions by enhancing individuals’ health consciousness. This suggested that mHealth usage recommendations from the external environment may encourage young adults’ attention to their health status, leading them to actively accept mHealth services.

Third, we discovered the moderating role of mHealth user experience in the complex relationship between social influence and behavioral intention. Specifically, the mHealth user experience negatively moderated the positive impact of social influence on trust (supported by H6c). This may be because young adults’ experiences as users of mHealth encourage them to trust it, diminishing the influence of external environmental information. Previous research has speculated that other trust bases can weaken or obscure certain trust bases [46]. Li et al. [53] discussed this in detail in a study on initial trust formation in organizational systems. They found that when cognitive reputation, calculus, organizational system trust bases, and social influence were simultaneously assessed, personality trust bases and technology system trust bases were overshadowed. Our results provide new evidence for this phenomenon.

Additionally, the experience of mHealth usage negatively moderated the positive impact of social influence on health consciousness (supported by H6b). This may be because health consciousness is based on the personal experiences of young adults who use mHealth. Social cognitive theory points out that the relationship between individuals, the environment, and behavior is not one-way but is instead dynamically interactive [67]. mHealth user behavior subjectively strengthens health consciousness and is, therefore, less sensitive to the influence of social recommendations. This study also found that the mHealth user experience positively moderated the positive impact of health consciousness on behavioral intentions (supported by H6e). This means that, compared with those without mHealth user experience, young adults who have used mHealth are more likely to be driven by health consciousness and demand mHealth for health management. Health behaviors driven by this need are more stable than herd mentalities [83].

### 4.2. Theoretical Implications

This study makes two theoretical contributions to the literature. First, our results verified the previous conjecture that there is more than one direct relationship between social influence and mHealth adoption intentions. The direct and indirect paths between social influence and behavioral intention were statistically significant, indicating that other variables partially mediated the relationship between social influence and behavioral intention. This study provides theoretical evidence that trust and health consciousness are mediating variables. This enriches the theoretical framework for predicting mHealth adoption behavioral intentions.

Second, we introduced mHealth user experience as a moderator in predicting mHealth behavioral intentions. The mHealth user experience significantly affected the relationship between other predictors and behavioral intention. These two groups may have viewed mHealth differently. This aspect was ignored in previous studies. Therefore, we recommend that future researchers adopt targeted measures that consider the characteristics of these two groups in their future work.

### 4.3. Practical Implications

This study is significant for the promotion and practice of mHealth. First, the positive effect of social influence shows that young adults can be effectively guided to actively adopt mHealth through promotions by people with great social influence. Therefore, developers and promoters using social media, word-of-mouth marketing, and other methods to increase positive social impact on target user groups may lead to higher mHealth adoption rates.

Second, trust partially mediated the relationship between social influence and behavioral intention. This suggests that trust building should be valued in improving behavioral intentions toward mHealth adoption. Therefore, developers can build and enhance user trust by strengthening privacy protection, providing accurate health information, and increasing transparency and user feedback mechanisms. At the same time, in addition to further improving the security policy for personal health data in mHealth applications and strengthening the supervision and protection of user data, the government can increase public awareness of mHealth by strengthening the promotion and education of mobile health applications, thereby enhancing the public’s trust in and willingness to use mobile medical care. Health consciousness partially mediated the relationship between social influence and behavioral intention. Therefore, in mHealth promotion activities, with the help of influential people, public health consciousness can be improved by strengthening health education and publicity, encouraging healthy behaviors and other publicity means, and guiding them to use mHealth applications more actively.

Third, the mHealth user experience was found to have a significant moderating effect on the model. This insight suggests that promotion strategies should differ between users who have already used the product and those who have not. For people without mHealth experience, their trust in mHealth-related information and increased self-health consciousness are more likely to be influenced by social influence. Therefore, in the early stages of mHealth promotion, strategies can be developed to expose more people to mHealth by establishing a positive social influence. For experienced mHealth users, their usage behavior is more likely to be driven by their health consciousness. Consequently, when designing health management applications, developers should consider ways to enhance users’ health consciousness to boost their motivation for continued use. These findings offer specific strategic guidance for the marketing of mHealth applications.

### 4.4. Limitations

Our study had several limitations. First, it only surveyed young adults at one university in China, and all participants were students. Given the specific nature of occupational identity, the results of this study should be interpreted with caution when interpreting the other occupational groups. Second, this study was conducted in China, and the country’s cultural background may have affected the results. The conclusions of this study should be verified in countries with different cultural backgrounds. Third, the data collection period for this study occurred after China issued policies to lift COVID-19 restrictions. People’s views on mHealth may have differed from those before the COVID-19 outbreak. This study lacks an in-depth exploration of the impact of the COVID-19 pandemic. In fact, during the COVID-19 pandemic, an increasing number of people began to contact and use mHealth proactively. Yang et al. [84] conducted a semi-longitudinal survey of 1593 mHealth applications. The results showed that, compared to before the pandemic, the downloads and usage of mHealth applications in China increased significantly after the pandemic. Therefore, in our future research, we intend to complete longitudinal investigations to explore the impact of COVID-19 on behavioral intentions to adopt mHealth.

## 5. Conclusions

Although mHealth has shown great potential in healthcare, it still faces many challenges in social popularization. This study uses the technology acceptance theory to examine how social influence affects behavioral intentions in mHealth adoption. Through surveys and analysis of a key population of young adults (18–40 years old), the research revealed several key findings. Our results highlight that trust and health consciousness play essential roles in the relationship between the two. Therefore, enhancing users’ trust in mHealth and health consciousness should be an important part of relevant promotion strategies. Additionally, we found a significant moderating effect of mHealth user experience in predicting mHealth behavioral intentions. For users who have mHealth user experience, their behavioral intentions are more likely to be influenced by health consciousness. This finding emphasizes that in product design, developers should pay more attention to improving users’ health awareness through mobile health services, thereby increasing their willingness to continue using mobile healthcare. These findings enrich the mHealth technology acceptance theory framework and provide an important theoretical basis for policymakers, product designers, and developers to formulate promotion strategies.

## Figures and Tables

**Figure 1 behavsci-14-00498-f001:**
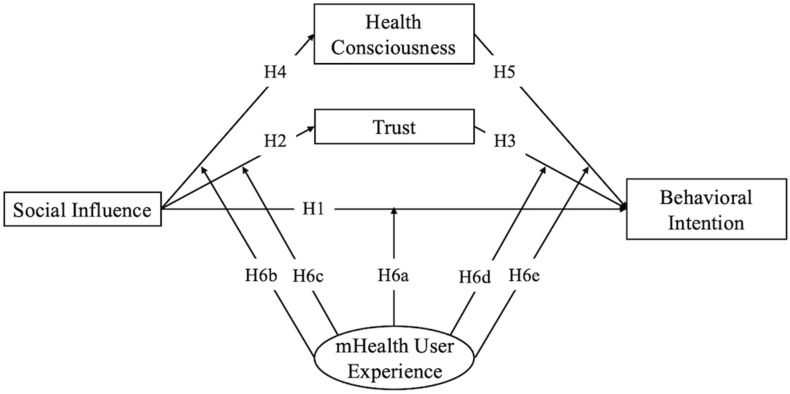
The research model proposed in this study. H: hypothesis.

**Figure 2 behavsci-14-00498-f002:**
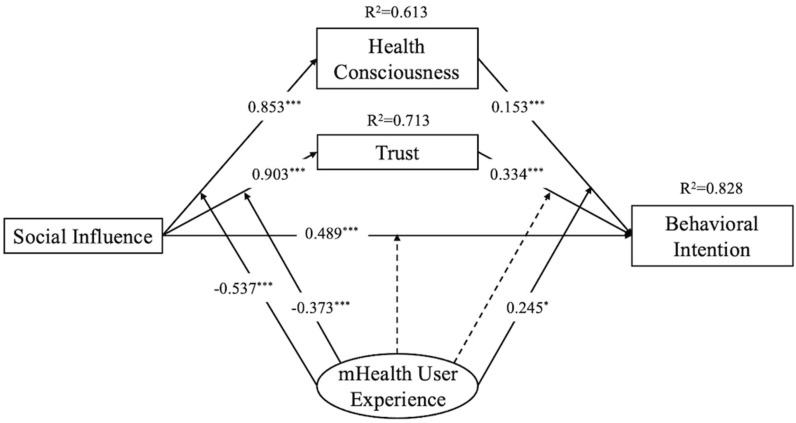
Final model results with moderator variable. * *p* < 0.05; *** *p* < 0.001.

**Table 1 behavsci-14-00498-t001:** Demographic characteristics of the respondents (N = 300).

Measure	Category	Value (N = 300), n (%)
**Age**		
	≤20	203 (67.7)
	21–30	93 (31)
	31–40	4 (1.3)
**Gender**		
	Men	109 (36.3)
	Women	191 (63.7)
**Job**		
	Student	300 (100)
**Smartphone Use (Year)**		
	1–3	23 (7.7)
	4–6	137 (45.7)
	8–10	89 (29.7)
	≥10	51 (17)
**mHealth User Experience**		
	Yes	67 (22.3)
	No	233 (77.7)

**Table 2 behavsci-14-00498-t002:** Outer loading test of questionnaire items.

Items	Item Loadings				*p*-Value
	BI ^a^	HC ^b^	SI ^c^	TR ^d^	
BI1	0.946				0.000
BI2	0.957				0.000
BI3	0.957				0.000
HC1		0.899			0.000
HC3		0.898			0.000
HC4		0.892			0.000
HC5		0.904			0.000
SI1			0.958		0.000
SI2			0.975		0.000
SI3			0.960		0.000
TR1				0.889	0.000
TR2				0.941	0.000
TR3				0.949	0.000
TR4				0.938	0.000
TR5				0.904	0.000

^a^ BI: behavioral intention. ^b^ HC: health consciousness. ^c^ SI: social influence. ^d^ TR: trust.

**Table 3 behavsci-14-00498-t003:** Validity and reliability tests.

Construct	Cronbach’s Alpha	rho_a	CR ^e^	AVE ^f^
BI ^a^	0.950	0.950	0.968	0.909
HC ^b^	0.920	0.924	0.943	0.807
SI ^c^	0.962	0.962	0.975	0.930
TR ^d^	0.957	0.958	0.967	0.854

^a^ BI: behavioral intention. ^b^ HC: health consciousness. ^c^ SI: social influence. ^d^ TR: trust. ^e^ CR: composite reliability. ^f^ AVE: average variance extracted.

**Table 4 behavsci-14-00498-t004:** Correlation matrix and the result of the Fornell and Larcker criterion tests.

Construct	Correlation			
	BI	HC	SI	TR
BI ^a^	0.953			
HC ^b^	0.793	0.898		
SI ^c^	0.858	0.748	0.964	
TR ^d^	0.853	0.776	0.830	0.924

^a^ BI: behavioral intention. ^b^ HC: health consciousness. ^c^ SI: social influence. ^d^ TR: trust.

**Table 5 behavsci-14-00498-t005:** Results of the hypotheses test.

Path (Hypothesis)	Model 1		Model 2		Hypothesis Test
	Path Coefficient	*p*-Value	Path Coefficient	*p*-Value	
SI ^a^ → BI ^b^ (H1)	0.407	0.000	0.489	0.000	Supported
SI → TR ^c^ (H2)	0.830	0.000	0.903	0.000	Supported
TR → BI (H3)	0.342	0.000	0.334	0.001	Supported
SI → HC ^d^ (H4)	0.748	0.000	0.853	0.000	Supported
HC → BI (H5)	0.226	0.000	0.153	0.030	Supported
mHealth User Experience × SI → BI (H6a)	N/A ^e^	N/A	−0.173	0.258	Unsupported
mHealth User Experience × SI → HC (H6b)	N/A	N/A	−0.537	0.000	Supported
mHealth User Experience × SI → TR (H6c)	N/A	N/A	−0.373	0.000	Supported
mHealth User Experience × TR → BI (H6d)	N/A	N/A	−0.004	0.979	Unsupported
mHealth User Experience × HC → BI (H6e)	N/A	N/A	0.245	0.040	Supported

^a^ SI: social influence. ^b^ BI: behavioral intention. ^c^ TR: trust. ^d^ HC: health consciousness. ^e^ N/A: Not applicable.

## Data Availability

Raw data were generated at Ritsumeikan University and Harbin Institute of Technology. Derived data supporting the findings of this study are available from the corresponding author S.Z. on request.

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
