# Peer review of "How Social Influence Promotes the Adoption of Mobile Health among Young Adults in China: A Systematic Analysis of Trust, Health Consciousness, and User Experience"

_behavsci, 2024, doi:10.3390/bs14060498_

Round 1

Reviewer 1 Report

Comments and Suggestions for Authors

This is a well-done study, and the writing could benefit from minor tweaking for the reader’s better understanding. 

From Line 53 to Line 77, the argument seems to start broad, then narrow to older adults, then discusses young adults, then discusses older adults, then discusses young adults to close the argument of the population of interest for the study, which somewhat confuses the reader of the subject and population of interest. Consider rearranging the sequencing of the argument so the reader can easily follow, from broad acceptance of mHealth and smart phone usage, that older adults are part of the population of consideration for adoption, but then framing the argument more narrowly to focus lastly on young adults as the link to adopt mHealth. 

Grammatical error Line 57.

There does not seem to be an operational definition for “young adults” age cut off from a reliable source, please provide it especially to justify that you included individuals up to age 40 years of age. 

Line 71: “they spend more time using the internet and have a higher penetration rate of smartphones…” Consider using a different word than penetration, I don’t think that the everyday reader would understand the phrasing of this. 

Section 1.1. is well-written.

Line 157 should read as plural hypotheses. 

In Section 2.2, it is important to list out the inclusion and exclusion criteria, and also the process of Informed Consent. Please also list the Ethics Board that provided the oversight for this study. 

For the Results section, it does not seem that the various H6 hypotheses are addressed.

For the Discussion section, could you please refer to the hypotheses that you are discussing, if there were findings or not, by the H.x. number. It’s difficult for a reader outside of your work to follow the information discussed. 

The Practical Implications section is well-written. 

Comments on the Quality of English Language

Minor errors.

Reviewer 2 Report

Comments and Suggestions for Authors

The article is written in a very clear way. Authors study how social influence affects young adults' intentions to adopt mobile health (mHealth) technologies in China. They are particularly interested in the roles of trust and health consciousness as mediating factors in this relationship. The article has a good structure and quality of English language. 

Statistical methods used should be described further. Please find below the given recommendations: 

1. Utaut should be explained when it first appears in the text.

2.UTAUT model proposed by Venkatesh et al. should be better explained before diving into numbers. 

3. To revise the arrangement on the page (line 332 and 333 should be on the same page).

4. After the lines 317-320 the authors should continue the explanation. For example ”meaning if people feel social pressure, they are more likely to intend to act accordingly. etc.” 

4.1. Same apply for lines 324 - 329, for each of the statistical conclusions.

5. I would change the phrase: ”Therefore, in future research, we recommend more longitudinal investigations to explore the impact of COVID-19 on behavioral intentions to adopt mHealth.” with ”Therefore, in our future research, we intend to.....”

6. The article reffered to trust, but did not describe the subject of issues of trust / challenges of mhealth, such as privacy, security, reliability etc.

7. Conclusions should be significantly extended. 
